# Shape deformation analysis reveals the temporal dynamics of cell-type-specific homeostatic and pathogenic responses to mutant huntingtin

Lucile Megret[1]*, Barbara Gris[2], Satish Sasidharan Nair[1], Jasmin Cevost[1], Mary Wertz[3], Jeff Aaronson[4], Jim Rosinski[4], Thomas F Vogt[4], Hilary Wilkinson[4], Myriam Heiman[3†], Christian Neri[1†]*

[1]Sorbonne Université, Centre National de la Recherche Scientifique UMR 8256, INSERM ERL U1164, Paris, France; [2]Sorbonne Université, Centre National de la Recherche Scientifique, Laboratoire Jacques-Louis Lyons (LJLL), Paris, France; [3]MIT, Broad Institute, MIT, Picower Institute for Learning and Memory, Cambridge, United States; [4]CHDI Foundation, Princeton, United States

**Abstract** Loss of cellular homeostasis has been implicated in the etiology of several neurodegenerative diseases (NDs). However, the molecular mechanisms that underlie this loss remain poorly understood on a systems level in each case. Here, using a novel computational approach to integrate dimensional RNA-seq and in vivo neuron survival data, we map the temporal dynamics of homeostatic and pathogenic responses in four striatal cell types of Huntington's disease (HD) model mice. This map shows that most pathogenic responses are mitigated and most homeostatic responses are decreased over time, suggesting that neuronal death in HD is primarily driven by the loss of homeostatic responses. Moreover, different cell types may lose similar homeostatic processes, for example, endosome biogenesis and mitochondrial quality control in *Drd1*-expressing neurons and astrocytes. HD relevance is validated by human stem cell, genome-wide association study, and post-mortem brain data. These findings provide a new paradigm and framework for therapeutic discovery in HD and other NDs.

*For correspondence:
lucile.megret@sorbonne-universite.fr (LM);
christian.neri@inserm.fr (CN)

†These authors contributed equally to this work

**Competing interests:** The authors declare that no competing interests exist.

## Introduction

Clinical studies of neurodegenerative disease (ND) progression, including that of Parkinson's and Huntington's disease (HD), provide evidence that progression to more advanced disease stages is often associated with a loss of compensatory processes that enable neural circuits to retain robust function even in the presence of some level of cellular dysfunction or loss (*Lee et al., 2000*; *Dubois et al., 2018*; *Gregory et al., 2018*). However, the dynamics of such changes remain poorly characterized on a molecular systems level. Precisely inferring how such biological processes are coordinated at a molecular systems level can be aided by the analysis of multidimensional datasets, for example, transcriptomic time-series data collected across genotypes or cell types. Such datasets are increasingly available to ND research (*Langfelder et al., 2016*; *Maniatis et al., 2019*), offering opportunities to comprehensively probe how molecular systems may respond to disease drivers, noticeably by using network inference (*Langfelder et al., 2016*; *Bigan et al., 2020*; *Mégret et al., 2020*). Probing molecular responses to disease drivers can greatly benefit from integrating transcriptomic data with functional screening data to discern pathogenic causative from compensatory responses. Here, we hypothesized that developing a new computational approach that utilizes an in-depth analysis of the shapes (e.g., surfaces, curves) that characterize genomic data will provide a

precise basis for mapping the dynamic and functional features of molecular responses on a systems level. To this end, we designed *Geomic*, an approach that is based on the shape deformation formalism (*Arguillère et al., 2015*). We applied this approach to the analysis of the cell-type-specific and temporal features of molecular responses in mouse models of Huntington's disease (HD), an ND that is caused by CAG repeat expansion in the huntingtin gene (*HTT*; *Zuccato et al., 2010*) and that is characterized by large changes to gene expression in the striatum and cortex (*Moumné et al., 2013*). Cell-type-specific studies are critical to understanding how each cell type responds to mutant *HTT* (mHTT) as, for example, striatal D2 dopamine receptor (*Drd2*)-expressing medium spiny neurons (MSNs) show enhanced vulnerability to the disease (*Reiner et al., 1988*), while some other striatal cell types are much less affected.

We used *Geomic* to integrate three datasets including (i) a multidimensional RNA-seq data characterizing gene expression changes in bulk striatal RNA of the allelic series of HD knock-in mice (Hdh mice), currently the largest RNA-seq reference dataset (six CAG repeat lengths: Q20 to Q175; three age points: 2 months, 6 months, 10 months), which was created to study how molecular responses may develop on a systems level in HD mouse models (*Langfelder et al., 2016*), (ii) a cell-type-specific RNA-seq dataset obtained from Hdh mice that were crossed with BAC-translating ribosome affinity purification (TRAP) mice (*Doyle et al., 2008*; *Heiman et al., 2008*) prior to cell-type-specific mRNA capture and sequencing, encompassing four striatal cell types and five CAG repeat lengths (20, 50, 111, 170, or 175 CAG repeats) at 6 months of age (*Lee et al., 2020*), and (iii) a neuron survival screening dataset obtained in the Hdh-Q175 mice upon infection of the striatum with genome-wide shRNA pools subcloned into lentivirus that preferentially transfect neurons (*Wertz et al., 2020*).

As described below, our approach identifies the temporal subtypes of pathogenic and compensatory responses, and the transitions thereof, as detected in *Drd1*-expressing MSNs, *Drd2*-expressing MSNs, striatal cholinergic (ChAT) interneurons, and striatal astrocytes, implicating stress response and cellular resilience pathways in all cell types. These responses involve genes that were previously shown to modulate HD pathogenesis in experimental models and/or to modify HD onset age, as well as genes previously unknown to modify disease course. A major feature of the *Geomic* map of molecular responses in the striatal cells of Hdh mice indicates that HD progression may be primarily driven by the loss of homeostatic responses, highlighting the alteration of similar homeostatic responses in different cell types, notably that of endosome biogenesis and mitochondrial quality control in *Drd1*-expressing neurons and striatal astrocytes. These data highlight that shape deformation analysis is an approach to help prioritize HD targets for prolonging compensatory responses and delaying HD progression in one or more cell types. These data also highlight the value of shape deformation analysis for accurate analysis of complex genomic datasets.

## Results

### Cell-type-specific assignment of whole-striatum gene deregulation surfaces in the Hdh model mice

To dissect the nature and evolution of the cell-type-specific molecular responses that may develop in the striatum of the 'allelic series' Hdh mice (a set of HD models that all possess a humanized exon one knock-in *Htt* allele and differ only by a series of increasing CAG repeat lengths; *Wheeler et al., 1999*; *Menalled et al., 2003*; *Heikkinen et al., 2012*; *Menalled et al., 2012*; *Langfelder et al., 2016*; *Franich et al., 2019*), we designed *Geomic*, a workflow in which the shape deformation principle is applied to the analysis of complex genomic data (*Figure 1*, see also Materials and methods). We used *Geomic* to integrate three datasets: (i) a multidimensional RNA-seq dataset characterizing gene expression changes in total RNA of the allelic series knock-in Hdh model mice, currently the largest reference dataset (six CAG repeat lengths: Q20 to Q175; three age points: 2 months, 6 months, 10 months), which was created to study how CAG length-dependent molecular responses to mHTT develop on a systems level in HD mouse models (*Langfelder et al., 2016*), (ii) a cell-type-specific RNA-seq dataset obtained from some of the same allelic series knock-in Hdh mice that were crossed with BAC-TRAP mice (*Doyle et al., 2008*; *Heiman et al., 2008*) prior to cell-type-specific mRNA capture and sequencing, encompassing four striatal cell types and several CAG repeat lengths (20, 50, 111, 170, 175 repeats) at 6 months of age (*Lee et al., 2020*), and (iii) an in vivo

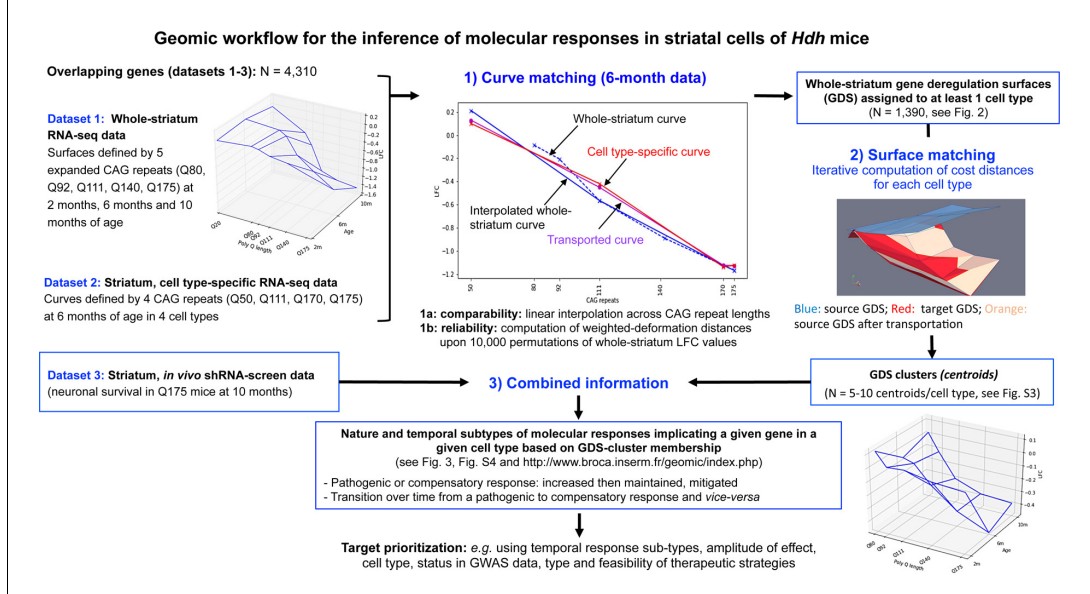

**Figure 1.** Application of shape deformation concepts to the detection of molecular responses in the striatum of Huntington's disease (HD) knock-in model mice. The *Geomic* protocol integrates three main steps for the integration of transcriptomic and cell survival data into a model that distinguishes the nature and temporal evolution of gene deregulation and molecular responses in a cell-type-specific manner. In step 2, a gene's cell type(s) expression is assigned to the bulk RNA-seq gene deregulation surface (GDS) by using the shape deformation cost for matching the data across expanding CAG repeats at the 6-month timepoint. It is noticeable that in many instances a gene that is downregulated across CAG repeats in a linear manner can be dysregulated across time in a non-linear manner, that is, increased then decreased expression (or vice versa), underlying the reduction of homeostatic responses or that of pathogenic responses over time. The data generated by *Geomic* analysis including the detection of cell types associated with whole-striatum gene deregulation, cell-type-specific GDS cluster centroids and identification of the type and temporal dynamics of molecular responses across cell types are available at http://www.broca.inserm.fr/geomic/index.php.

The online version of this article includes the following figure supplement(s) for figure 1:

**Figure supplement 1.** Shape deformation principle applied to the comparison of curves defined by genomic data.
**Figure supplement 2.** Cumulative distribution of the deformation distances between gene deregulation curves in the striatum of Hdh model mice.
**Figure supplement 3.** Centroids of the gene deregulation surface clusters associated with specific cell types in the striatum of Hdh model mice.
**Figure supplement 4.** Validation studies of the cost distance for clustering gene expression surfaces.

neuronal survival genetic screening dataset that identified genome-wide modifiers of striatal neuron survival in the context of mHtt (*Wertz et al., 2020*).

We focused our *Geomic* analysis on a group of 4310 genes of high interest, that is, those for which data on whole-striatum gene expression deregulation (*Langfelder et al., 2016*), striatal cell-type-specific gene expression (*Lee et al., 2020*), and effect on striatal neuron survival upon in vivo shRNA screen (*Wertz et al., 2020*) were available in the Hdh mice. As a first step of *Geomic* analysis (*Figure 1*), we compared the whole-striatum gene deregulation curves defined by the evolution of the log2-fold-change (LFC) across expanded CAG repeats at the 6 months of age model timepoint – which is part of the gene deregulation surface (GDS) defined by RNA-seq data across 15 CAG repeats and three age points in the striatum of Hdh mice (*Langfelder et al., 2016*; see http://www.broca.inserm.fr/geomic/index.php; the database might take some time to load at first consultation) – and striatal cell-type-specific curves defined by the evolution of gene expression changes (LFC) across four expanded CAG repeats. A linear interpolation on the data was made in order to make the curves comparable. We then used a metric, namely the deformation distance, based on the cost of deformation of the interpolated whole-striatum curves to the cell-type-specific curves (see Materials and methods). With this approach, a total of 1390 whole-striatum GDS could be mapped at high confidence to at least one cell type, including 747 GDS mapped to *Drd1*-MSNs, 167 GDS mapped to *Drd2*-MSNs, 156 GDS mapped to striatal ChAT cholinergic interneurons, and 321 GDS mapped to striatal astrocytes (*Figure 2*, *Supplementary file 1*). The whole-striatum deregulation of 542 genes was specifically mapped to *Drd1*-MSNs, that of 91 genes specifically mapped to *Drd2*-MSNs, that of 87 genes specifically mapped to ChAT cholinergic interneurons, and that of 185 genes

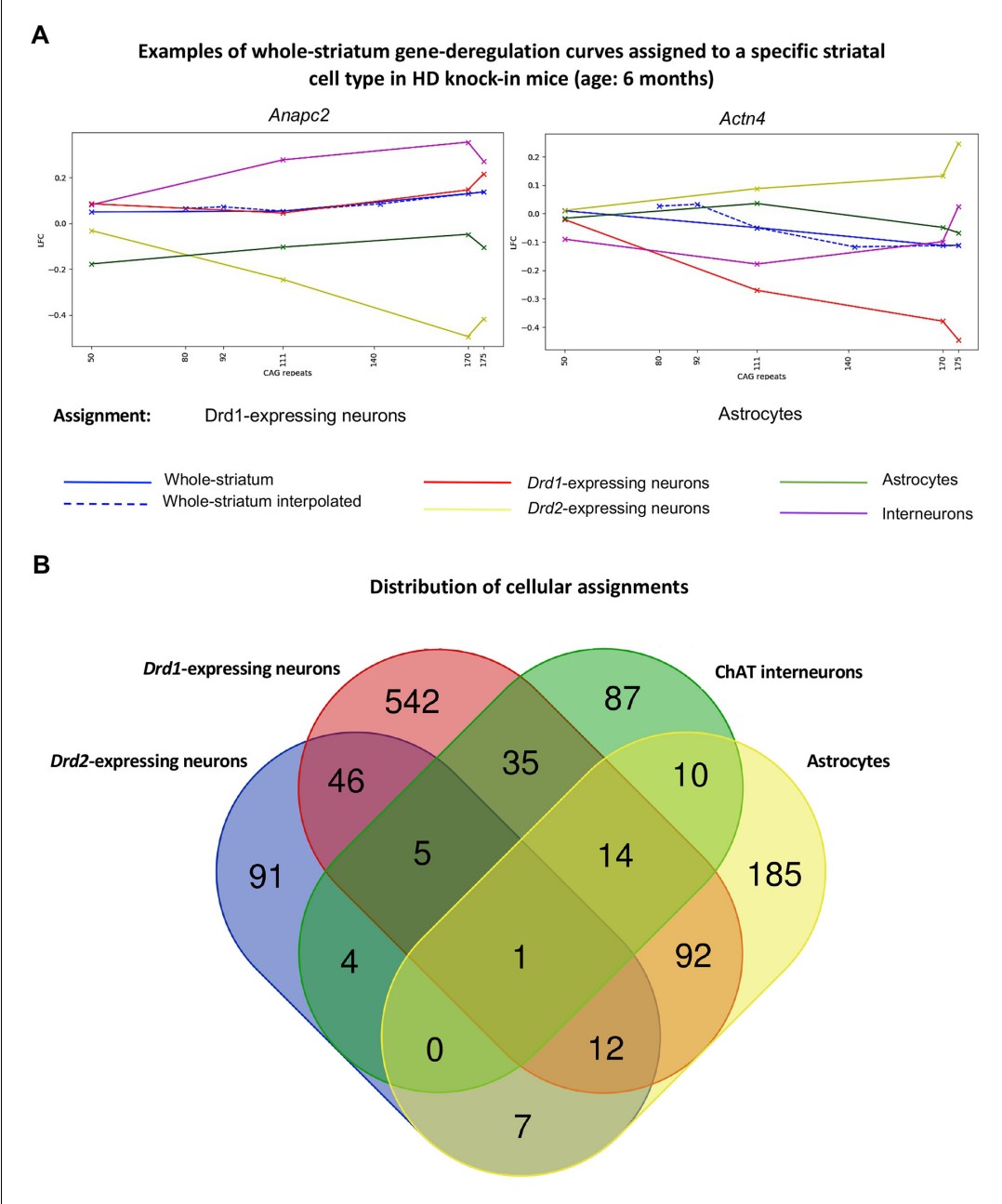

**Figure 2.** Mapping of whole-striatum gene deregulation surfaces to striatal cell types in Huntington's disease (HD) knock-in model mice. *Geomic* analysis of whole-striatum and cell-type-specific RNA-seq data using gene deregulation curves at 6 months of age (see *Figure 1*) attributed a total of 1390 whole-striatum gene deregulation surfaces to at least one striatal cell type (see *Supplementary file 2*). (**A**) Examples of genes for which whole-striatum deregulation was specifically mapped to *Drd1*-expressing neurons or to astrocytes (see also http://www.broca.inserm.fr/geomic/index.php). (**B**) Distribution of cellular assignment. Most expression surfaces recapitulating gene deregulation across CAG repeats and age points in the striatum of Hdh model mice were assigned to one cell type. Cellular assignment suggests that 64 genes may be similarly deregulated across CAG repeat lengths in *Drd2*- and *Drd1*-expressing neurons and that 111 genes may be similarly deregulated across CAG repeat lengths in astrocytes and either *Drd2*- or *Drd1*-expressing neurons. The Venn diagram was generated using the Venn tool at http://bioinformatics.psb.ugent.be/webtools/Venn/.

specifically mapped to astrocytes (*Figure 2*). The difference in the number of these assignments between the MSN populations reflects the greater CAG length-dependent transcriptional dysregulation in *Drd1*-MSNs (when analyzed by linear regression across allele lengths; *Lee et al., 2020*), suggesting that this phenomenon is inherent to the CAG repeat-dependent dynamics of whole-striatum gene deregulation data.

## Surface deformation analysis identifies precise clusters of gene deregulation across striatal cell types in the Hdh mice

Next, we used shape deformation concepts to identify the prototypical classes of GDS that may be associated with a given cell type and to define the pathways and biological processes that may be primarily associated with each GDS cluster. To this end, we used the cost distance between GDS to compute the GDS clusters summarized by the centroid of each cluster, for each cell type (*Figure 1*/ step 2 'Surface matching', *Figure supplement 2*; see also Materials and methods). In some cases, the whole-striatum deregulation of some genes could be assigned to more than one striatal cell type, signifying that such genes may participate into more than one GDS cluster. This analysis identified 5–10 GDS clusters and their corresponding centroids *per* cell type, mostly in *Drd1*-MSNs and astrocytes (*Figure 1— figure supplement 3*, *Supplementary file 2*). Interestingly, biological content analysis (see Materials and methods), including KEGG pathway and gene ontology (GO) analysis, revealed that, in addition to implicating the 'HD pathway' (in interneurons, astrocytes) and pathways involved in neuronal activity (e.g., 'Wnt signaling' and 'cAMP signaling' in *Drd2*-MSNs), GDS clusters implicate stress response pathways in all cell types (*Supplementary file 2*). Noticeably, cluster centroid Drd2-1 (*Drd2*-MSNs: cluster centroid 1, down-regulation across CAG repeats) implicates stress response and cell survival genes such as *Hipk4*, a kinase that promotes HD pathogenesis in transgenic flies (*Al-Ramahi et al., 2018*), and *Arpp21*, a cAMP-regulated phosphoprotein that protects neurons against HD pathogenesis in transgenic *Caenorhabditis elegans* nematodes (*Bigan et al., 2020*). Using weighted-edge network analysis of the whole-striatum time-series RNA-seq data, we previously found that these two genes belong to a dynamic network that implicates cell survival and cellular senescence and that forms in the striatum of Hdh mice as they become highly symptomatic on a behavioral level (i.e., gene nodes come together into short-path interactions at 10 months of age; *Bigan et al., 2020*), which, as defined herein, may primarily occur in *Drd2*-MSNs. Cluster centroids Drd2-5 (up-regulation), Drd1-6 (up-regulation), Drd1-8 (down-regulation), Interneurons-0 (down-regulation), Interneurons-4, and Astrocytes-4 (up-regulation) implicate 'DNA repair' or 'DNA damage response', suggesting that DNA repair is a biological process that is significantly altered in HD models in all four cell types. Cluster centroid Drd1-1 (down-regulation) implicates 'DNA damage' and 'inflammation', and cluster centroid Astrocytes-4 (up-regulation) implicates 'cytosolic-DNA sensing' pathway, suggesting that the response to cytosolic DNA upon cellular insult may be altered in HD in both MSNs and astroglia. Cluster centroid Drd1-0 (down-regulation) implicates 'FOXO signaling' and 'cellular senescence', processes which are also implicated by cluster centroid Drd2-1. Of note, the same pathway (or same biological process) displaying different directionality of gene expression change in different cell types is an expected pattern due to the specificities of gene deregulation in different cell types. Regarding the same cell type, the same pathway displaying dysregulated genes with different directionality of change (which is rarely observed, e.g., for the pathway amphetamine addiction in *Drd2*-MSNs) or the same biological process displaying such a pattern of dysregulated genes (which is more frequent) are also expected as, for example, genes of opposing function in the same pathway or biological process would be expected to change in opposite directions. Collectively, these results indicate that the biological space in which we examine the dynamics molecular responses to mHTT in the striatum of Hdh mice – via the analysis of 4310 informative genes – includes the regulation of neuronal activity and that of the cell survival/death balance. The latter system may be essential for neuronal and non-neuronal cells to resist neurodegenerative states. These results provide the first shape deformation resource and thus a precise basis for studying how CAG repeat length in mHTT may alter gene expression and striatal biology in an age- and cell-type-dependent manner (http://www.broca.inserm.fr/geomic/index.php).

## Integrating shape deformation data and functional data identifies the temporal subtypes of molecular responses across striatal cell types in the Hdh mice

Having characterized the cell-type-specific features of gene deregulation via shape deformation analysis, we used this information to interrogate the dynamics of the molecular responses that may operate in a cell-type-specific manner in the striatum of Hdh mice (*Figure 1*, step 3). To this end, we combined information from GDS cluster centroids (involving 1131 genes and 1390 cellular assignments) with that from the in vivo shRNA viability screen (4310 genes tested) based on the effect

(protective, pathogenic: 793 genes) of a gene knockdown on neuron survival data in the Q175 mice as inferred from the effect (suppressor, enhancer) of the corresponding shRNA as tested at 10 months of age (see Materials and methods). Among the 4310 genes tested in this viability screen (*Wertz et al., 2020*), 3517 genes showed no effect, among which 1149 genes with a cellular assignment (see http://www.broca.inserm.fr/geomic/index.php). For 241 genes, a conclusive assessment on both the viability effect (enhancer, suppressor) and cellular assignment could be made. Specifically, we identified two temporal subtypes of molecular responses that in each cell type may develop in the striatum of the *z*Q175DN mice, including two temporal subtypes (increased then maintained, increased then reduced) of either compensatory or pathogenic responses, and two types of response inversions, including transition from pathogenicity to compensation and vice versa (*Supplementary file 3*; see also http://www.broca.inserm.fr/geomic/index.php). More specifically, we found 103 genes that underlie a compensatory response, including 56 genes in *Drd1*-MSNs, 10 genes in *Drd2*-MSNs, 11 genes in ChAT cholinergic interneuron, and 23 genes in astrocytes (*Supplementary file 3*). In contrast, we identified 136 genes that underlie a pathogenic response, including 75 genes in *Drd1*-MSNs, 18 genes in *Drd2*-MSNs, 12 genes in ChAT cholinergic interneurons, and 31 genes in astrocytes (*Supplementary file 3*). Interestingly, these data show that (i) the majority of pathogenic responses are reduced over time in *Drd1*-MSNs (73%), *Drd2*-MSNs (83%), and astrocytes (64%), and (ii) the majority of compensatory responses are reduced over time in *Drd1*-MSNs (69%), *Drd2*-MSNs (80%), and astrocytes (69%; *Supplementary file 3*3). Additionally, these data show that strong reduction is a more frequent event in the compensatory response group (34%) compared to the pathogenic response group (25%) as observed for *Drd1*-MSNs, *Drd2*-MSNs, and astrocytes (*Supplementary file 3*). These features suggest that neuronal decline and death in HD is primarily caused by the loss of compensatory responses over time, and not by the increase in strength of pathogenic mechanisms. These data also show that reduction of a response may actually evolve to a transition from pathogenicity to compensation (or the reverse), a phenomenon that is observed in astrocytes and associated with the inversion of deregulation of three genes. Importantly, network analysis and biological annotations (see Materials and methods) suggested that each class of molecular responses is biologically homogeneous, containing both distinct and shared biological features in each cell type compared to the other ones (see http://www.broca.inserm.fr/geomic/index.php). Notably, *Rab7* and *Clasp1*, both HTT protein interactors, are retained in the class 'compensatory response/increased then reduced overtime' for *Drd1*-MSNs (*Figure 3*) and astrocytes (*Figure 3—figure supplement 1*), highlighting alteration of endosome biogenesis and mitochondrial quality control as common disease drivers in these cells. The importance of the progressive loss of homeostatic responses is also highlighted by network convergence analysis of the increased-pathogenic and reduced-compensatory responses (i.e., first-degree neighbors common to both types of molecular responses and genes, e.g., 143 genes in *Drd1*-MSNs). This analysis identified clusters of signaling systems that may be primarily affected by HD over time, highlighting not only proteasome-mediated protein catabolism ($p=5.93 \times 10^{-62}$) but also regulation of cell cycle ($p=2.47 \times 10^{-5}$: e.g., G1 and G2 transitions), senescence-associated secretory phenotypes ($p=8.32 \times 10^{-17}$), DNA repair ($p=9.01 \times 10^{-5}$), TGF-ß signaling ($p=0.0019$), and *Rab*-dependent vesicle trafficking (e.g., *Rab1*, *Rac1*, *Gdi1*, *Chm*) as primary targets of the combined gain of pathogenicity and loss of compensation in *Drd1*-MSNs.

Collectively, these results suggest that stress response and cellular resilience pathways are strongly affected in the response to mHtt on a systems level, emphasizing the re-instigation of those compensatory responses that are strongly reduced over time in *Drd1*-MSNs, *Drd2*-MSNs, and astrocytes as important approaches to counteract disease evolution.

## Disease relevance of genes implicated in the molecular responses defined by *Geomic* analysis

The genome-wide in vivo neuronal survival data obtained in *z*Q175DN mice (*Wertz et al., 2020*) provide strong evidence for the genes involved in these effects to be causally involved in HD pathogenesis. To further investigate this aspect, we analyzed the relevance to available data regarding causal gene effects in HD pathogenesis. We performed this analysis for the genes implicated in the molecular responses defined at the cell type level (group I) and a larger group involving both these group I genes and the genes that modulate neuronal survival but for which cellular assignment is not available (group II genes).

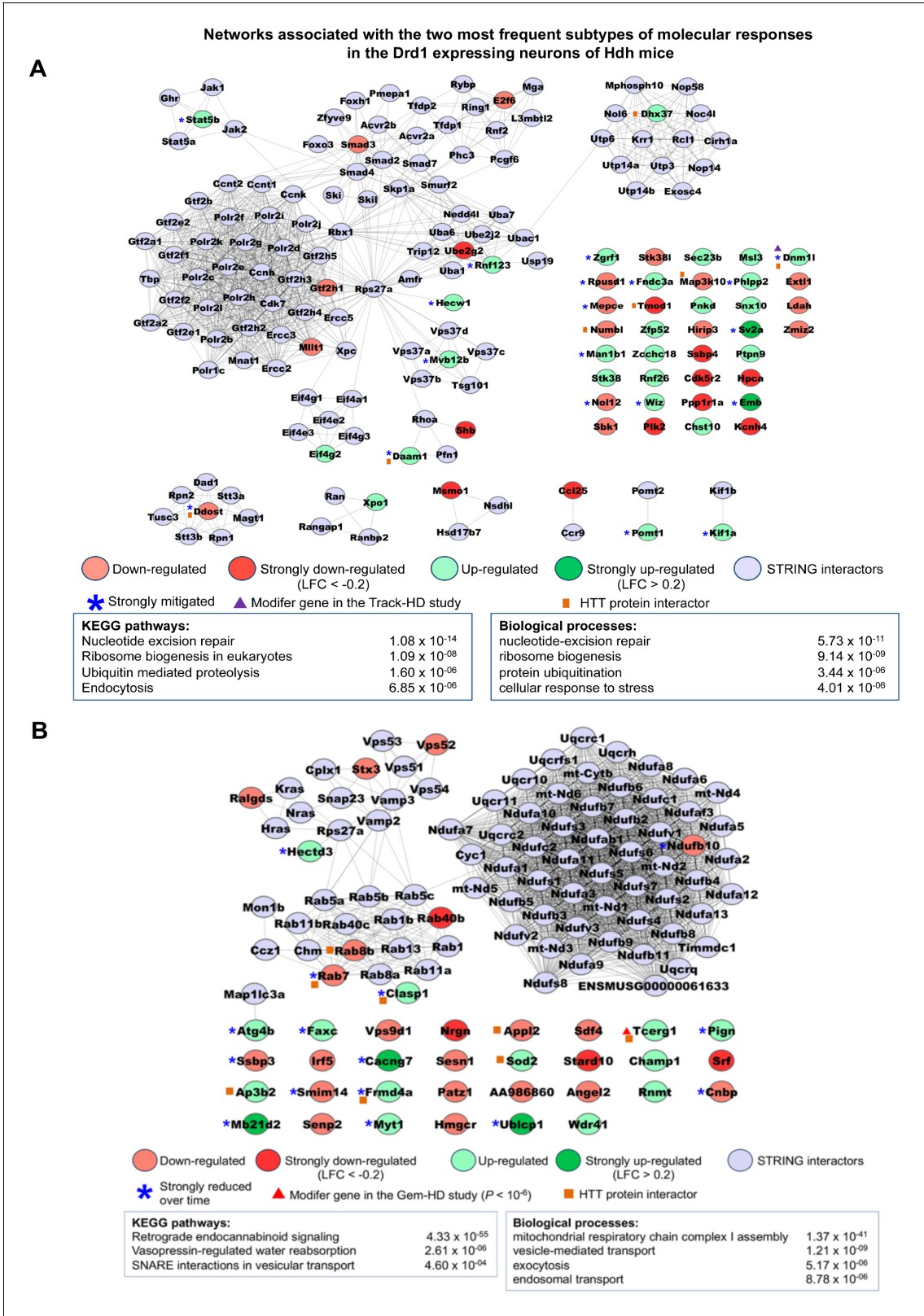

**Figure 3.** Networks associated with the most frequent temporal subtypes of molecular responses in the *Drd1*-expressing neurons of Huntington's disease (HD) knock-in model mice. Network representations of the most frequent subtypes (i.e., increased-then-reduced responses) of molecular responses that developed in *Drd1*-expressing neurons in the striatum of Hdh model mice. The genes retained in each of these subtypes of molecular responses (see *Supplementary file 3*) were used as seeds to build networks comprising high-confidence neighbors added on the first shell as inferred

*Figure 3 continued*

from the STRING database (see Materials and methods). Top 2–4 enrichments for biological annotations (KEGG pathways, gene ontology [GO] biological processes) are shown where top annotations have the smallest p-values for the largest numbers of genes as indicated by the STRING database. The networks for all subtypes of molecular responses developed by *Drd1*-expressing neurons can also be seen at http://www.broca.inserm.fr/geomic/index.php. (**A**) Pathogenic responses that are mitigated over time (*n* = 55 seed genes). (**B**) Compensatory responses that are increased then reduced (*n* = 39 seed genes).

The online version of this article includes the following figure supplement(s) for figure 3:

**Figure supplement 1.** Networks associated with the most frequent temporal subtypes of molecular responses in the astrocytes of Hdh model mice.

First, we identified nine mouse genes that in group I or group II are conserved in Drosophila and that modify climbing in transgenic flies with pan-neuronal expression of human *HTT* species (*Al-Ramahi et al., 2018*; *Figure 4A*). These overlaps do not reach significance. Of note, the down-regulation of some of these genes (*Cacna1b*, *Drd2*, *Pcmt1*) is protective in both *Htt* flies and Hdh model mice, whereas that of some other genes (*Plk2*, *Actn1*, *Prepl*) shows opposite effects in Hdh model mice and in *Htt* flies (e.g., down-regulation of Plk2 is pathogenic, promoting neuronal death in Hdh model mice, but it is a compensatory response promoting climbing in *Htt* flies), raising the possibility that the same gene expression change may differently impact on neuronal circuit function versus neuronal survival. Interestingly, we identified significant overlaps for up to 34 mouse genes that are conserved in *C. elegans* and that modify the response to light touch (RNAi screen) in transgenic nematodes with expression of human N-terminal *HTT* in touch receptor neurons (*Lejeune et al., 2012*), including nine group I genes (p=0.043) and 34 group II genes (p=5 $\times$ 10$^{-4}$; *Figure 4B*). About 45% of these overlapping genes showed a similar effect on touch response in *Htt* nematodes (*Lejeune et al., 2012*) and neuronal survival in Hdh mice (*Wertz et al., 2020*) upon knockdown, suggesting that while the same gene expression change could differently impact on neuronal function compared to neuronal survival, a significant proportion of gene expression changes may similarly impact on both phenotypes.

Second, we tested for overlap with SNP data relevant to human gene orthologs that modify early-stage human disease in a total of 1991 human HD carriers (the TRACK-HD study), as measured using a composite progression score (*Moss et al., 2017*) or that modify the age-at-onset of motor symptoms in over 9000 human HD patients (*Genetic Modifiers of Huntington's Disease (GeM-HD) Consortium, 2019*). One gene, *Dnm1l* (pathogenic response assigned to astrocytes and to *Drd1*-MSNs and *Drd2*-MSNs), a dynamin-1-like protein and GTPase that regulates mitochondrial fission, is in overlap with the TRACK-HD study. We also identified six genes that are implicated in a molecular response in the zQ175DN mice and that have a human ortholog harboring an HD onset modifier with a genomic significance comprised between 10$^{-6}$ and 10$^{-4}$, including *Golga4*, a golgin, *Tcerg1*, a transcription elongation regulator (also known as CA150), *Farp2*, an ARH/RhoGEF gene also associated with Parkinson's disease and *Bag1*, a chaperone regulator important for the proteasome and lysosome in group I, extending to *Msh3*, a mismatch repair protein, and *Hdlbp*, a high-density lipoprotein binding protein, in group II (*Figure 4C*). Noticeably, we previously identified *TCERG1* as a partner protein of huntingtin and a polymorphic repeat DNA (encoding an imperfect (Gln-Ala)(38) tract) in *TCERG1* as a potential modifier of HD onset (*Holbert et al., 2001*) whose expression rescues striatal cell death in lentiviral overexpression (rats) and knock-in (mouse cells) models of mutant huntingtin neurotoxicity (*Arango et al., 2006*). Here, *Geomic* analysis indicates that up-regulation of *Tcerg1* may correspond to a compensatory response that tends to increase over time in the zQ175DN, in *Drd1*-MSNs (http://www.broca.inserm.fr/geomic/). In *C. elegans*, *tcer-1*/TCERG1 may promote longevity upon germline removal, acting in association with *daf-16*/FOXO (*McCormick et al., 2012*). The *daf-16*/FOXO gene strongly protects against neuronal dysfunction in 128Q nematodes, and FOXO3 protects mouse striatal cells derived from HD knock-in mice from cellular vulnerability (*Tourette et al., 2014*). Thus, the *Geomic* analysis indicates that targeted manipulation of *TCERG1* in *Drd1*-MSNs, potentially along with its direct or indirect interactors, holds promise to delay the HD-associated pathogenic mechanisms. The value of the dynamic molecular response map provided by the *Geomic* analysis for precise target prioritization is also illustrated by the observation that astrocyte-directed inhibition of *Golga4* is of highest priority, as this golgin is up- and then down-regulated in astrocytes, promoting a pathogenic response that turns

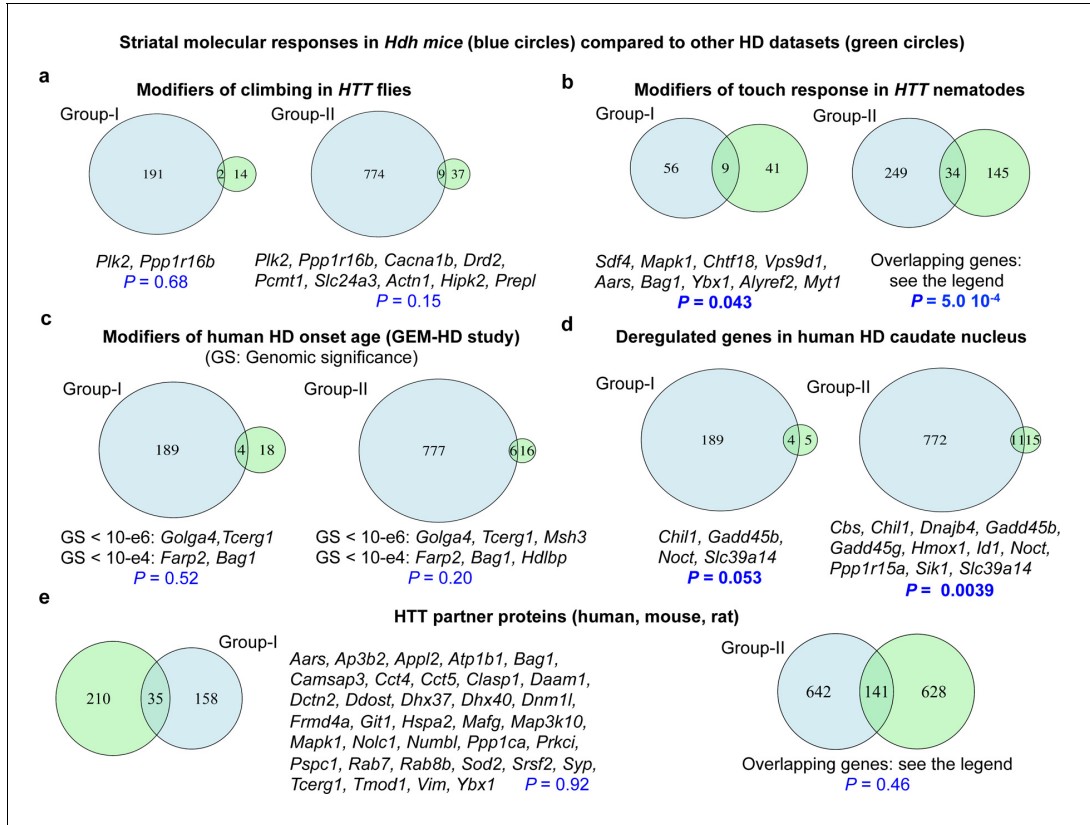

**Figure 4.** Comparison of molecular responses in the striatum of Hdh model mice to Huntington's disease (HD) datasets. Overlaps between the molecular responses identified by *Geomic* analysis in the striatum of Hdh model mice and other HD datasets. We performed these comparisons for the genes implicated in molecular responses defined at the cellular level (group I) and a larger group (group II) involving the union of group I genes and the genes that modulate neuronal survival but for which cellular assignment is not available (see Materials and methods). (A) Overlaps with genes that are conserved in Drosophila and that modify climbing in transgenic flies with pan-neuronal expression of human Htt species (*Al-Ramahi et al., 2018*). (B) Overlaps with genes that are conserved in *Caenorhabditis elegans* and that modify light-touch response in transgenic nematodes with expression of human N-terminal HTT in touch receptor neurons (*Lejeune et al., 2012*). Genes in overlap with group II are *L3mbtl2, Slc17a6, Bche, Ugp2, Col25a1, Sdf4, Mapk1, Ccne1, Kctd3, Gabbr1, Slc23a2, Atp6v0a1, Ip6k2, Snw1, Chtf18, Pax6, Pak3, Pak1, Ltbp4, Snrpb, Vps9d1, Aars, Bag1, Lta4h, Tpo, Ybx1, Cope, Alyref2, Kbtbd4, Plscr3, Tbce, Nhp2, Uqcc1,* and *Myt1*. (C) Overlaps with genes that are conserved in humans and that are associated with the modification of age at motor onset of HD ($p<10^{-4}$) in the GEM-HD participants (*Genetic Modifiers of Huntington's Disease (GeM-HD) Consortium, 2019*). (D) Overlaps with genes that are conserved in humans and that are deregulated in the caudate nucleus of HD patients ($N = 2$) compared to control participants (*Agus et al., 2019*). (E) Overlaps with established or putative HTT protein interactors, here those identified using human, mouse, and rat libraries (see the HINT resource at https://chdifoundation.org/hdinhd/). HTT protein interactors in overlap with group II are *Aars, Abi2, Actn1, Actr2, Adgrg1, Akr7a5, Alg2, Ank2, Ap3b1, Ap3b2, Appl2, Arpc5, Ascc3, Atp1b1, Atp6v0a1, Atp6v1b2, Bag1, Bin1, Cacna1b, Camk2a, Camk2g, Camsap3, Cbs, Cct4, Cct5, Clasp1, Cldn11, Copa, Coro2a, Csnk2b, Cyc1, Daam1, Dapk1, Dctn1, Dctn2, Ddost, Dhx37, Dhx40, Dnajc4, Dnm1l, Drd2, Eef2, Ehbp1, Eno3, Epb41, Erap1, Fahd2a, Fbl, Fhl2, Fis1, Flot2, Frmd4a, Galnt13, Git1, Gpd2, Grm3, Hdac8, Hdlbp, Hsp90b1, Hspa2, Hspa8, Kars, Kif5c, Kpna2, Lyar, Mafg, Man1a2, Map1lc3a, Map3k10, Map6, Map7d1, Mapk1, Mbd4, Med14, Msh3, Myo1b, Nckap1, Ndufaf4, Ndufv1, Nedd4, Nf1, Nhp2, Nolc1, Nos1, Numbl, Pabpc1, Pabpn1, Pacsin3, Pak1, Pard3, Pde4b, Pex11b, Pex5l, Picalm, Pip5k1c, Pop4, Ppil3, Ppp1ca, Ppp1r12c, Prdx2, Prkci, Pspc1, Rab7, Rab8b, Rbm39, Rgs4, Rpl10a, Rps19, Rps6ka2, Rps6ka5, Rpsa, Satb1, Scarb2, Setdb1, Sfxn5, Slc17a6, Snd1, Snrpb, Snw1, Sod2, Sqstm1, Srrm1, Syp, Syt12, Taok1, Tcerg1, Tecr, Tmod1, Tpm3, Tram1, Ubtf, Uso1, Utp15, Vdac3, Vim, Vsnl1, Wdr12, Wrnip1, Ybx1, Zdbf2,* and *Zfp169*.

into a compensatory response as the zQ175DN model mice develop strong behavioral symptoms (http://www.broca.inserm.fr/geomic/index.php). Golgi stress responses have been previously associated with HD (*Sbodio et al., 2018*), and golgins are important for vesicle-mediated transport and for retrograde transport to the endoplasmic reticulum, the latter involving huntingtin (*Brandstaetter et al., 2014*). Thus, targeting the down-regulation of Golga4 in astrocytes could mitigate HD progression.

Third, we identified significant overlaps between either group I ($p<0.053$) or group II ($p<0.0039$) genes and the human orthologs dysregulated in the caudate nucleus of human prodromal HD brains

(*Agus et al., 2019*; *Figure 4D*). Although Agus et al. investigated gene deregulation in two prodromal HD brains, this overlap suggests that molecular responses in the striatum of Hdh mice may be relevant to molecular pathology of HD in the human caudate. We also identified overlaps between group I (*n* = 35) and group II (*n* = 141) genes and the HDinHD list of established and putative HTT protein interactors, herein restrained to those interactors found using human, mouse, and rat libraries (*Figure 4E*; see also http://www.broca.inserm.fr/geomic/index.php). Although not statistically significant, these overlaps involved several genes, suggesting that both HTT-proximal and non-HTT-proximal pathways are important for the molecular responses to mHTT in the striatum of Hdh model mice.

Finally, we performed a literature review to seek prior evidence for genes that are involved in the molecular responses identified by our *Geomic* analysis as modulating HD-associated pathogenesis, as defined by neuronal dysfunction phenotypes across invertebrate, cellular, and murine models of the disease. Here, we restricted this analysis to the group I genes that show a strong level of dysregulation (LFC>0.5) in the *z*Q175DN mice across age points, which includes six genes (*Penk*, *Htr1b*, *Hpca*, *Chtf18*, *Kcnh4*, *Ppp1r1a*) implicated in a pathogenic response as developed in at least one cell type and five genes (*Ppp1ca*, *Foxo1*, *Ppp1r16b*, *Kcng1*, *Nfe2l3*) implicated in a compensatory response (see *Supplementary file 3*). Among the genes implicated in a pathogenic response, the stress response and proenkephalin *Penk* (down-regulation assigned to *Drd2*-MSNs: pathogenic response that is aggravated then maintained) is a gene whose overexpression improves disease symptoms in transgenic (R6/2) HD mice (*Bissonnette et al., 2013*). The 5-hydroxytryptamine (serotonin) receptor *Htr1b* (down-regulation assigned to *Drd2*-MSNs: pathogenic response that is aggravated then maintained) has been strongly associated with NDs, including HD (*Pang et al., 2009*). The DNA polymerase *Chtf18* (up-regulation assigned to *Drd1*-MSNs: pathogenic response that is aggravated then maintained) may be involved in DNA repair, a process genetically associated with human HD onset age (*Genetic Modifiers of Huntington's Disease (GeM-HD) Consortium, 2019*). Voltage-gated potassium channels such as *Kcnh4* (down-regulation assigned to *Drd1*-MSNs: pathogenic response that is mitigated) may be associated with motor deficits in transgenic and knock-in HD model mice (*Sebastianutto et al., 2017*). Finally, although protein phosphatase one regulatory inhibitor subunit 1A *Ppp1r1a* (down-regulation assigned to *Drd1*-MSNs: pathogenic response that is mitigated) is not formally linked to neurobehavioral phenotypes in HD models, protein phosphatase inhibition may be relevant to several diseases, including NDs (*Vintonyak et al., 2011*). Among the genes implicated in a compensatory response, *Ppp1ca* (down-regulation assigned to *Drd2*-MSNs: compensatory response that is increased then maintained) and *Ppp1r16b* (down-regulation assigned to *Drd2*-MSNs: compensatory response that is increased then maintained) are protein phosphatases that like *Ppp1r1a* may be relevant to NDs (*Vintonyak et al., 2011*). The gene *Foxo1* (down-regulation assigned to *Drd2*-MSNs: compensatory response that is increased then maintained) belongs to the Forkhead family of transcriptions factors, a class of stress response factors that is associated with modulation of neuronal dysfunction in NDs, including HD, as shown in *C. elegans* (*daf-16/*FOXO) and cellular (*Foxo3*) models (*Parker et al., 2005*; *Tourette et al., 2014*). The gene *Kcng1* (up-regulation assigned to *Drd1*-MSNs: compensatory response that is increased then maintained) is a voltage-gated potassium channel that, like *Kcnh4*, may be associated with motor deficits in transgenic and knock-in HD model mice (*Sebastianutto et al., 2017*). Finally, the transcription and stress-response factor N*fe2l3* (up-regulation assigned to *Drd1*-MSNs and astrocytes: compensatory response that increased then maintained) might be relevant to HD in that this protein acts via multiple mechanisms (e.g., UPR, protein quality control of the ER, inflammation, cell division) that are associated with HD pathogenesis (*Chowdhury et al., 2017*).

Together, these results indicate that several of the molecular responses defined herein to modulate neuronal survival in the striatum of Hdh mice may also modulate neuronal activity in HD models and/or gene expression or age-at-onset in human HD.

## Discussion

Highly dimensional genomic datasets (*Langfelder et al., 2016*; *Maniatis et al., 2019*; *Mégret et al., 2020*) offer the possibility of dissecting the complexity of biological processes on a molecular level, for example, the context-dependent features of molecular regulation dynamics. Here, we report the first application of the shape deformation formalism to the analysis of omics data. Our results

suggest that this approach precisely identifies the cell-type-specific and temporal features of molecular responses to mHTT in the mouse brain.

To ensure reliability and accuracy in mapping cell-type-specific and temporal features of molecular responses to mHTT, we used a shape deformation analysis workflow that is completely data-driven. Reliability was ensured by permutation analysis for the key analytical step of the workflow used herein, namely gene deregulation curve matching (see *Figure 1*, step 1). With regard to accuracy, our analysis is rather stringent as suggested by comparing the data-driven threshold (which ensures that a distance is not small by chance) and the cumulative distribution features of the deformation distances between gene deregulation curves (*Figure 1—figure supplement 2*). High stringency relates to the fact that the probability of making a hypothesis by chance is estimated by the number of times that a hypothesis is retrieved after permutations. Since many LFCs may be similar across data layers (see http://www.broca.inserm.fr/geomic/index.php), closeness between curves may often remain almost identical upon permutations, leading to an overestimate of the amount of times a hypothesis is obtained by chance. Part of this effect (high stringency of data-driven threshold) is intrinsic to the input RNA-seq data in which some of the CAG repeat data points in the cell-type-specific data are not exactly the same as compared to the whole-striatum data, requiring linear interpolation to ensure curve comparability prior to performing curve matching analysis (see *Figure 1*, Step 1).

Another important feature of our shape deformation analysis is biological accountability and analytical versatility. Prior to defining the nature and temporal dynamics of the molecular responses in a cell-type-dependent manner, a key step of the shape deformation analysis used herein is to calculate a shape deformation distance (between curves) in order to assign whole-striatum gene deregulation to a given cell type(s), followed by calculating a shape deformation distance (between surfaces) to classify the whole-tissue GDS in a shape- and cell-type-dependent manner (*Figure 1*). The resulting GDS clusters may be viewed as blocks of genes that are under the same type of transcriptional control in the cell type to which they are assigned. However, we did not design curve matching to detect cancellation effects — that is, gene deregulation that evolves in an opposite manner in two (or more) cell types, which may result in whole-tissue (here the mouse striatum) gene deregulation shapes (curves, surfaces) that tend to remain flat across conditions — as detecting cancellation effects does not enable a conclusion to be reached regarding the cellular assignment(s) of gene deregulation and the dynamics of the molecular response that is associated with gene deregulation. Yet, cancellation effects could arise from a compensatory response in one cell type that is accompanied by a pathogenic response in another cell type, and the use of shape deformation concepts in the *Geomic* workflow can be adapted to the detection of such events, as for example observed for 4833420G17Rik, Adgrg1, and 1700025G04Rik (see http://www.broca.inserm.fr/geomic/index.php). Along the same lines, whole-tissue gene deregulation may reflect a wide range of cell-type- and gene-specific phenomena such as the deregulation of a gene that is mostly expressed in one cell type, gene deregulation in the most abundant cell type(s) in that tissue (which in the mouse striatum are *Drd1*-MSNs and *Drd2*-MSNs for neurons, and astrocytes and oligodendrocytes for glia), and a strong level of gene deregulation in low abundance cell types (e.g., ChAT interneurons) that is not mitigated by the expression level of that gene in more abundant cell populations. The curve matching analysis performed herein readily accounts for these phenomena as they will all decrease the deformation distance between the whole-striatum gene deregulation curve and the curve(s) corresponding to the cell type(s) in which gene deregulation mostly happens. The formula for calculating the deformation distance thus includes no a priori knowledge of the relationships between the shape of gene expression dysregulation across CAG repeats and the abundance of striatal cell populations. Yet, the use of shape deformation concepts can be adapted to formally account for the influence of parameters such as cell type abundance or cell-type-specific mRNA abundance, for example, by adjusting the calculation of the cost for deforming a shape into another one.

Shape deformation analysis may be able to reduce data inconsistency across data points. Data inconsistency may relate to variations of raw data quality or to threshold effects specific to a given condition (e.g., cell type, timepoint). In the striatum of Hdh mice, RNA-seq data quality is highly homogeneous (*Langfelder et al., 2016*; *Lee et al., 2020*). However, the level of some cell-type-specific gene deregulation in these mice can vary between Q170 and Q175, particularly in glial cells, suggesting threshold effects in the response of these cells. The design of the deformation distance is able to address this problem. We reduced the weight associated with the change of LFC between

these two data points compared to the changes across the other CAG repeat lengths (see Materials and methods), minimizing the loss of information that could be associated with such abrupt effects without impairing the power of *Geomic* analysis of molecular responses across cell types.

Building up onto the richness and high dimensionality of genomic data collected in the striatum of Hdh model mice (*Langfelder et al., 2016*; *Lee et al., 2020*; *Wertz et al., 2020*), our *Geomic* analysis is notably able to detect a temporal subtype (increased then maintained) of pathogenic and compensatory responses that, in each cell type, may evolve in a mostly monotonic manner as Hdh model mice become increasingly symptomatic on behavioral levels. Strikingly, this *Geomic* analysis reveals that the majority of pathogenic and compensatory responses may be reduced over time, including significant proportions (25–35%) that are strongly reduced, particularly in the compensatory response group (*Supplementary file 3*). Moreover, this *Geomic* analysis also indicates that in rare instances a molecular response may be lost over time, with transitions from a pathogenic to a compensatory response (and vice versa) detected in striatal astrocytes as the Hdh model mice become increasingly symptomatic. Compared to less-detailed genome-scale information – for example, protective or pathogenic gene in a single cell type (*Lejeune et al., 2012*), compensatory or pathogenic responses based on a single timepoint (*Al-Ramahi et al., 2018*) – and besides expected rules for molecular responses to unfold such as linear increase over time, *Geomic* information on the cell-type- and disease phase-dependent features of molecular responses in HD knock-in mice identifies two unexpected rules for these responses to unfold, including (i) both pathogenic and compensatory responses, but particularly compensatory responses, can be frequently and strongly reduced as Hdh model mice become increasingly symptomatic, suggesting that loss of compensatory responses is a main driver of disease progression, and (ii) molecular responses can become inverted over time (although this is not frequent). Enrichr analysis (*Chen et al., 2013*) highlighted no significant differences between the transcription factors associated with molecular responses that increase and those associated with molecular responses that are increased then reduced over time. Although tools such as Enrichr may lack precision for striatal cell types, we speculate that these rules reflect the temporal dynamics of chromatin remodeling in response to HD in the mouse striatum (*Achour et al., 2015*). Uncovering these two unexpected rules enhances precision with target prioritization in HD as it is relevant to enriching the modalities for therapeutic intervention (i.e., prolonging compensation, neuronal versus glial targeting), allowing specific groups of targets (e.g., up-regulated genes implicated in a compensatory response that decreases over time) to be found. The main rule identified in our study, that is, neuronal death induced by mutant huntingtin may be primarily driven by the loss of homeostatic responses, is noticeably relevant to the loss of homeostatic responses such as the activation of autophagy (see, e.g., *Atg4b*; *Figure 3B*), which corroborates, on a molecular systems level, the importance of autophagy in HD as previously supported by the role of HTT as a scaffold protein in autophagy (*Ochaba et al., 2014*; *Rui et al., 2015*; *Croce and Yamamoto, 2019*). Of particular interest are the genes that may be knocked down in a cell-type-specific manner to avoid reduction of compensatory responses (particularly those that are strongly decreased over time; see, for instance, *Bora*, *Cdk17*, *Rab10b*, *Cnbp*, and *Rab7* at http://www.broca.inserm.fr/geomic/index.php). This level of biological precision is particularly useful for target prioritization in the context of increasing knowledge on the pre-symptomatic/prodromal phases of HD (*Moss et al., 2017*) as a potential treatment window, on the genetic modifiers of HD (*Genetic Modifiers of Huntington's Disease (GeM-HD) Consortium, 2019*), and on gene deregulation (RNA-seq data; *Agus et al., 2019*) and cell population remodeling (single-nucleus RNA-seq data; *Al-Dalahmah et al., 2020*; *Lee et al., 2020*) in post-mortem HD brains.

On a global level, our *Geomic* analysis captured cell-type-specific features from whole-striatum gene expression data for a large number of the 4310 genes initially analyzed, whether or not they were significantly dysregulated in the Hdh model mice bearing large CAG repeat expansions (Q170–Q175). Concordant and complementary information was obtained in the original analysis of the TRAP-seq data. However, in several instances, the dysregulation of certain genes was not assigned by the *Geomic* analysis to the same cell type that dysregulation was noted in the original analysis of the TRAP-seq data. These discrepancies included genes that were significantly dysregulated in striatal cell types of the zQ175DN model mice. These discrepancies are not primarily attributable to the strong difference in gene expression levels that may be observed for the zQ175DN allele compared to smaller knock-in CAG repeat alleles as this phenomenon is not systematically observed, can vary

across striatal cell types, and is minimized by the design of the *Geomic* analysis (see above; see also Materials and methods/weighted cost distance). Rather, these discrepancies may be attributable to the logic underlying the striatal cell type assignment(s) of whole-striatum gene expression dysregulation in *Geomic*, which is based on matching gene expression curves, across CAG repeat lengths, using a threshold that ensures robust conclusions as, for every gene, using this threshold tends to retain the cell-type-specific curve that is most closely similar to the whole-striatum curve, excluding the other cell-type-specific curves even though they show some shape similarity with the whole striatum curve.

With regard to the signaling pathways highlighted by *Geomic* analysis, our results suggest that surface deformation analysis can be used to efficiently cluster GDS into prototypical classes of gene deregulation that are small in size and biologically homogeneous. GDS clusters contain 2–135 surfaces (or genes) with a median value of about 35 genes per cluster and a biologically homogeneous content (see *Supplementary file 2*), suggesting that the biological precision of shape deformation analysis is very high. Notably, GDS cluster data highlighted the alteration of processes associated with neuronal activity and synaptic transmission, as expected, also highlighting the alteration of specific stress response and cellular resilience mechanisms such as DNA damage repair and mitochondrial homeostasis. The precision of GSD cluster data in turn fosters precise conclusions about the signaling pathways and biological processes involved in each class of molecular response across cell types. In astrocytes, mHTT may up-regulate cytosolic DNA sensing pathways, raising the possibility that release of nucleic acids in the cytosol of astrocytes might be associated with HD pathogenesis, as recently validated for the release of mitochondrial RNA in the MSNs of Hdh mice (*Lee et al., 2020*). The development of cellular senescence in human HD neural stem cells and MSNs (*Voisin et al., 2020*) provides an additional level of validation for the central importance of the dynamics of stress response in HD. *Geomic* analysis provided detailed information on the nature and evolution of molecular response to HD in the mouse striatum mostly for *Drd1*-MSNs (direct pathway striatal projection neurons) and astrocytes. Since *Drd2*-MSNs (indirect pathway striatal projection neurons) are mostly vulnerable to HD in patients (*Reiner et al., 1988*), this observation suggests that *Drd2*-MSNs are unable to activate a proper compensatory response and to ensure successful cellular maintenance and survival. The proportion of compensatory changes is rather similar in both cell types (42% in *Drd1*-MSNs, 35% in *Drd2*-MSNs), which also applies to pathogenic changes in *Drd2*-MSNs (65%) compared to *Drd1*-MSNs (58%) neurons, and both cell types appear to engage similar cell compensation mechanisms (e.g., vesicular trafficking, TOR signaling, FOXO signaling, autophagy; see http://www.broca.inserm.fr/geomic/index.php). Thus, the smaller number ($n$ = 10 in *Drd2*-MSNs, $n$ = 56 in *Drd1*-MSNs) and weaker efficacy (through decrease over time) of compensatory responses in *Drd2*-MSNs versus *Drd1*-MSNs support this possibility. Noticeably, TOR signaling and regulation of autophagy are engaged by the compensatory responses that are increased then maintained in *Drd1*-MSNs, whereas they are engaged by those that are increased then reduced in *Drd2*-MSNs. Additionally, our *Drd1*-MSNs data may document the consequences of the loss of corticostriatal terminals, which may mostly affect *Drd1*-MSNs in Q140 knock-in mice prior to overt motor symptoms (*Deng et al., 2014*).

In summary, our data identify the temporal dynamics of cell-type-specific compensatory and pathogenic responses for regulation of neuronal survival in the striatum of HD knock-in mice, providing a blueprint to select therapeutic targets for re-instating neuronal resilience and to select biomarker(s) for monitoring whether candidate drugs may engage homeostatic mechanisms for efficacy. Our data show that using shape deformation analysis of complex genomic data can precisely uncover the context-dependent features of molecular responses, highlighting the applicability of this approach for biological accuracy in modeling high dimensional genomic data. Our data notably identifies previously undetected molecular patterns and rules that, in specific cell types, may underlie the decreased capacity of neural circuits to cope with the HD process over time. This information provides a novel framework to understand, in a mammalian context, how (e.g., through gene knockdown) and when to re-instigate cellular compensation and to oppose the HD process, as instructed by HD model mice.

## Materials and methods

### Removal of outliers in expression data

To gain homogeneity in RNA-seq data, we constructed an Euclidean-distance sample network for each age point in the whole-striatum RNA-seq data obtained in the allelic series of Hdh model mice (*Langfelder et al., 2016*) and for each CAG repeat length in the cell-type-specific (*Drd1*-neurons, *Drd2*-neurons, ChAT cholinergic interneurons, and astrocytes) snRNA-seq data from TRAP mice (*Lee et al., 2020*). We then removed those mouse samples whose standardized inter-sample connectivity Z.k was below a threshold set to 2.5.

### Differential expression analysis

To identify deregulated genes in either the whole striatum (*Langfelder et al., 2016*) or specific striatal cell types (this study) of Hdh model mice, significant read count (>10 counts in at least one of the samples) data for eight individuals (four males and four females) as available in the RNA-seq data in the allelic series of Hdh model mice was fed into Deseq2 implemented in the R package DESeq2 (version 3.8 on R 3.5.2; *Love et al., 2014*) in order to obtain an LFC vector for each condition (CAG repeat length >20, age) and a vector indicating if the genes are significantly deregulated (p<0.05) or unchanged (p≥0.05) for each condition. The set age $k$, Q20 is used as reference for each condition at age $k$ and Q>20. In this series, we have 10 individuals for each condition (males). We applied the same analysis to the cell-type-specific gene deregulation data (*Lee et al., 2020*).

### Selection of genes for modeling the dynamics of molecular responses

We retained only the genes that have a significant LFC at 6 months of age in the whole striatum for, at least, two conditions. We further retained only those which are in the cell-type-specific LFC tables (regardless of the significance of the LFC value in those tables) and that were tested for effect of gene knockdown (shRNA screen) on neuron survival in the striatum of Q175 mice at 10 months of age is available. This analysis retained a total of 4310 genes.

### Deformation distances and cost distances for shape deformation analysis

Previous studies have suggested that two major phenomena may underlie molecular pathology in HD, particularly in the striatum, including (i) a positive correlation between CAG repeat length in Htt and the strength of gene deregulation (which also applies to the number of deregulated genes), and (ii) a linear increase or decrease of gene expression levels over time, particularly for the genes that are strongly deregulated by HD (*Langfelder et al., 2016*). One may thus assume that, for example, a gene that is strongly associated to HD may become increasingly deregulated across CAG repeat lengths (progression). Another example of such a progression is a phenomenon of diffusion of the deregulation across conditions so that, if a gene is up-regulated at 6 months, this gene may remain up-regulated at 10 months if no additional perturbation is involved. To compare the deregulation profiles of two genes using shape deformation analysis, we thus gave a similar importance to the dynamics of the deregulation and to the amplitude of that deregulation. To this end, we considered the LFC values as the points of a discretized shape (curve, surface). This approach allows the shape deformation formalism (*Arguillère et al., 2015*) to be used in order to quantify the differences between the deregulation patterns of two genes by calculating how difficult it is to deform a gene deregulation shape into another one. More specifically, we applied a simplified version of the deformation module analysis (*Gris et al., 2018*), which allows to make use of prior knowledge in the definition of the distances, as developed below.

To test for similarity between the shapes (curves, surfaces) defined by genomic data, we developed the following analysis. We observe a series of points $(S_i)_{1 \leq i \leq N}$ (discretization of curves or surfaces). We quantified the difference between two series of points $(S_i)$ and $(T_i)$ via the difficulty to deform the series of points $S_i$ in order to bring it as close as possible to the series of points $T_i$. The allowed deformations are constrained and are parameterized by some vectors $h_i$, herein referred to as the control vector, carried by each of the points $S_i$. For each couple of points $i$ and $j$, we define a weight $a_{ij}$ specifying how the point $j$ acts on the point $i$. The idea underlying this approach is that if $S_j$ moves of $h_j$ then, as a natural and cost-less consequence, $S_i$ will also move of $a_{ij}h_j$. The set of

weights is a modeling choice, based on biological hypotheses. Note that if $a_i$ is equal to zero, then the point $j$ does not act on $i$. Given this weight, a deformation parameterized by the vector $h=(h_i)_{1\leq i\leq N}$ transforms the points $S_j$ into points $S_j'$ such that $S_i' = S_i \sum_{j=1}^N a_{i,j}h_j$. This transformation may be seen as a simplification of the deformation module principle (*Gris et al., 2018*).

For a shape $S$, we note $X^S=(X_i^S)_{1\leq i\leq N}$ the vector of $\mathbb{R}^N$ that is formed by the LFC of the points of $S$. Let us pose $A = (a_i) \in \mathbb{R}^N\times\mathbb{R}^N$, the action matrix.

A control vector $h$ is an element of $\mathbb{R}^N$ (where $N$ is the number of points that defines each shape) and transforms a source shape $S$ into a new shape $S'$ defined by $X^{S'}=X^S+Ah$.

We now define the deformation distance between a source shape $S$ and a target shape $T$ by

$$d(S,T) = \min_{h\in\mathbb{R}^N}\mu\|h\|^2+\|T-(S+Ah)\|^2.$$

This function is a sum of the deformation cost $\mu\|h\|^2$ associated to a control vector $h$ and the mismatch quantity which is the Euclidean distance between the target and the transported source. The positive parameter $\mu$ allows the importance of the deformation cost to be weighted, which can be set by the user. A simple calculation shows that the optimal control vector $h_S$ exists and is explicitly given by

$$h_{S,T} = (\mu Id + A^tA)^{-1}A^t(T-S). \tag{1}$$

Thus, the deformation distance between a source shape $S$ and a target shape $T$ is given by

$$d(S,T) = \mu\|h_{S,T}\|^2+\|T-(S+Ah_{S,T})\|^2.$$

We define the matrix $D$ as to:

$$D = (\mu Id + A^tA)^{-1}A^t.$$

Here, the matrix $A$ is a matrix that contains only positive terms. The matrix $D$ is well defined as, if $\mu Id+A^tA$ is not invertible, then $-\mu$ would be an eigenvalue of $A^tA$, which implies that, with $X$ an associated eigenvector,

$$0> -\mu|X|^2 = -\mu<X> =' AX,X> =,AX> = |AX|^2 \geq 0.$$

With the explicit expression of $h_S$, we can verify that

$$d(S,T) = <\mu D^tD + (Id-AD)^t(Id-AD)(T-S),T-S>. \tag{2}$$

Note that $d$ defines a real distance as if $(S,T)=0$, then from its definition we get $h_{S,T}=0$. As a consequence, since $h_S = D(S-T)$ with $D$ invertible, $S = T$.

We refer to $d$ as the deformation distance. We can also consider a derived distance defined by

$$d(S,T) = |h_{S,T}|^2 = |D(S-T)|^2, \tag{3}$$

herein defined as the cost distance. The cost distance only takes into account the cost associated with transforming $S$ into a shape that is similar as much as possible to $T$ (with the allowed deformations), whereas the deformation distance also considers the mismatch term.

We specify below the matrices $A$ that we use in our analysis. As indicated above, our analysis is based on a uniform evolution of a measure (e.g., gene deregulation) across a set of variables to match or to not match shapes. To this end, we consider that a point (LFC value for a given condition, e.g., CAG repeat length or age) acts on the higher condition (same CAG repeat length and older age, or same age and longer CAG repeat length) and we calculate the cost associated to map a source shape to a target shape (*Figure 1—figure supplement 1*). To test the benefit of the cost term in the definition of the distance, we compared the performance of the cost distance with that of other types of distances as detailed further below (see Validation of the cost distance).

## Assignment of whole-striatum gene deregulation to striatal cell types via curve matching

### Definition of action matrices

In Hdh model mice, the LFC values at 6 months of age in the whole-striatum and cell-type-specific gene deregulation data are the discretization points of curves. In this case, each point is defined by an LFC value for a given condition (e.g., a CAG repeat length). We set $a_i=1$ if $j \leq i$ (higher CAG repeat length), $a_{i,j}=0$ otherwise. This action matrix reduces the weight of the LFC change between Q170 and Q175. For every gene, we compute the deformation distance between whole-striatum and cell-type-specific gene deregulation curves. If the deformation distance between these two curves is smaller than the threshold (see 'Cellular assignment of whole-striatum gene deregulation' for the determination of threshold), then a cellular assignment is retained. Whole-striatum gene deregulation can be assigned to zero, one, or several (up to four) cell types.

### Linear interpolation

The whole-striatum gene deregulation data and the cell-type-specific data in Hdh mice are not fully comparable because the CAG repeat data points are not exactly the same between these two datasets. To make these two datasets fully comparable, we used linear interpolation to transform CAG repeat data points in the whole-striatum gene deregulation curves at 6 months into CAG repeat data points as available in the cell-type-specific gene deregulation curves. This interpolation was not applied to the Q111 data point as it is common to the two datasets. For Q170, we took the value

$$LF(\text{Q140}) + (\text{LFC}(\text{Q175}) - \text{LFC}(\text{Q140}))\frac{170 - 140}{175 - 140}$$

which corresponds to the intersection of the axis $x$ = Q170 and the segments connecting the LFC at Q140 and Q175 in the whole striatum. For Q50, we take the value

$$LF(\text{Q80})\frac{92 - 50}{92 - 80} - \text{LFC}(\text{Q90})\frac{80 - 50}{92 - 80}$$

which corresponds to the intersection of the axis $x$ = Q50 and the prolongation of the segments connecting LFC at Q80 and at Q92 in the whole striatum.

### Calculation of deformation distances

To calculate the deformation distances, we transformed each one table of LFC values (whole striatum data, cell-type-specific data) into two tables of transformed LFCs. The first table is obtained by multiplying the original LFC by the matrix $D = (\mu Id + A^t A^{-1})^t$, as defined in *Equation (1)*. The second table is obtained by multiplying the original LFC values by the matrix $Id - AD$. Given *Equation (2)*, the deformation distance between two curves is equal to the weighted sum of the Euclidean distance between the transformed LFC values.

### Calculation of weighted deformation distance

For each gene, we compute the deformation distance between the interpolated whole-striatum gene deregulation curve and each striatal cell-type-specific gene deregulation curve in Hdh model mice at 6 months of age. Here, a confounding factor is that curves with a high amplitude may be further far away from each other than the ones with a small amplitude, which is true even though the resulting distances are small compared to amplitudes. To overcome this problem, we transformed the deformation distance (see above) into a weighted deformation distance that takes into account the LFC amplitude (maximum LFC value across CAG repeats). The weighted deformation distance between the curve for gene $G$ in the whole-striatum data and the curve for the gene $G$ in a given cell-type-specific dataset is computed as:

$$100 \times \frac{\mathrm{d}\big(\text{G}_{striatal}, \text{G}_{cell\text{-}type}\big)}{0.5 + \min\big(amp(\text{G}_{striatal}), amp\big(\text{G}_{cell\text{-}type}\big)\big)},$$

where $d(\text{G}_{striatal}, \text{G}_{cell\text{-}type})$ is the deformation distance between $A$ and $B$, $amp(\text{G}_{striatal})$ is the maximum

LFC of $G$ in the whole-striatum data, and $amp(G_{cell\text{-}type})$ is the maximum LFC of $G$ in the cell-type-specific data.

## Cellular assignment of whole-striatum gene deregulation via curve deformation analysis

Whole-striatum GDS were assigned to a specific cell type through the comparison of gene deregulation curves at 6 months of age, computing the deformation distances and applying a threshold that is determined in a data-driven manner. The difficulty is to select the threshold based on which we may conclude on shape similarity. In other words, we do not know what the typical distance may be between a whole-striatum gene deregulation curve and a cell-type-specific gene deregulation curve. We herein cannot rely on using the cumulative distribution to select a threshold as a confounding factor can be that whole-striatum deregulation is very close to a cell-type-specific deregulation and as this may happen for many genes. This situation might then lead to selecting a threshold that is typical of any one distance between two curves in the data space and that could miss too many true hypotheses or retain too many false hypotheses. We thus reasoned that a better approach is to select the threshold so that a maximum of hypotheses (gene deregulation mapped to a cell type), obtained by using this threshold, does not survive permutations. To this end, we tested several thresholds from the minimum up to the third quantile (integers) of the distribution of distance values, here ranging from 2 to 23 (*Figure 1—figure supplement 2*). We then selected the one threshold that shows the highest number of assignments with a false discovery rate (FDR) < 0.1, as detailed below.

To estimate the number of assignments that may be obtained by chance for a given threshold, we computed deformation distances upon 10,000 permutations of the LFC values across the five expanded CAG repeat lengths in the whole-striatum gene deregulation data.

Of note, permutating only the curves between genes to perform permutations is not enough as several whole-striatum LFC curves are very close to each other (which is expected assuming that the expression of several genes may be controlled by the same block of transcriptional modulators). We thus independently permuted the LFC for each CAG repeat length between genes. The p-value for each hypothesis on a threshold is given by the number of trials for which this hypothesis is obtained using this threshold. We then used the vector of p-values associated to each threshold to compute an FDR using the Benjamini and Hochberg method (Benjamini Y, 1995) and the function p.adjust of the package StatsModels (https://www.statsmodels.org/0.8.0/#) and statsmodels.stats written in Python. The FDR is a method for conceptualizing the rate of false positives when conducting multiple comparisons. We used FDR as FDR-controlling procedures are considered to be more powerful than the Bonferroni's correction for multiple testing. Finally, we retained the threshold that maximizes the number of assignments with q-value <0.1, that is, a threshold value of 8. This approach enabled the whole-striatum deregulation of 1390 genes out of the 4310 genes initially considered to be mapped to a cell type. As illustrated by the cumulative distribution of the entire set of deformation distances and by gene deregulation curves (see http://www.broca.inserm.fr/geomic/), this threshold is rather stringent (*Figure 1—figure supplement 2*).

## Construction of gene deregulation surface clusters for each cell type

### Definition of action matrices

The LFC values for the three age points and five CAG repeat lengths in the whole striatum are the discretization points of surfaces. In this case, each point is defined by an LFC value for a condition (an age point and a CAG repeat length). We set $a_i=1$ if the point $i$ is a higher condition (older and same CAG repeat length, or same age and longer CAG repeat) than the point, $a_{i,j}=0$ otherwise. We analyzed the dynamics of gene expression changes with a cellular assignment, independently for each cell type. More precisely, to identify the prototypical patterns of gene deregulation, we clustered the gene deregulation surfaces associated to each cell type. To this end, we applied a K-means algorithm as implemented in sklearn.cluster in python using the cost distance. More precisely, the K-means algorithm was applied to the tables of transformed LFCs, obtained by multiplying the original LFCs by the matrix $(\mu Id+A^tA^{-1})^t$, as defined in *Equation (3)*. This analysis allowed us to use the classical K-means algorithm to be used in a simple way as the cost distance on original LFC is equal to the Euclidean distance on the transformed LFC.

One important parameter when running a K-means algorithm is the number of clusters. Here, this number is chosen independently for each cell type, based on the evolution of the sum of squared errors and the Rand index (*Rand, 1971*) computed using the function adjustedrandscore of sklearn. cluster. Running the algorithm 10 times, we verified that all series of clusters have an adjusted rand score higher than 0.75, which ensures robustness. We performed one clustering analysis *per* cell type, using all gene deregulation effects that were mapped to at least one cell type, meaning that a given gene may be recruited into clusters for different cell types. The centroids obtained for each GDS cluster using the K-means algorithm are shown in *Figure 1—figure supplement 3*.

## Validation of cost distance for clustering gene deregulation surfaces

We used the cost distance defined above to cluster gene deregulation surfaces. However, one may argue that other distances could be used such as, for example, the Euclidean distance (a widely used metric), or the correlation distance (see below the formula). Here, we show that using the Euclidean distance and the correlation distance may perform sub-optimally for shape comparison compared to using the cost distance. The advantage of the cost distance is to take into account the direction of an effect across conditions, without neglecting the amplitude of this effect. Using two prototypical examples, each based on three simulation profiles (SPs) made of toy surfaces as developed hereafter, we show that the Euclidean distance fails to properly take into account the direction of an effect and that the correlation distance gives too much weight to the noise.

The first simulation is devoted to test the superiority of the cost distance over the Euclidean distance (*Figure 1—figure supplement 4*; simulation 1). In this simulation, the reference surface in the SPs is built in such a way that there is no variation of LFC values across CAG repeats. In the reference surfaces for all SPs, the LFC value is equal to 0 at 2 months of age. The LFC value is equal to 1 at 6 months and at 10 months for SP1, to 1 at 6 months and to 0 at 10 months for SP2, and to $(\sqrt{2}-1)/\sqrt{2}$ at 6 and 10 months for SP3. We then generate 50 surfaces around each reference surface and SP by adding a random perturbation (0,0.1). SP1 is thus most similar to SP3 than to SP2 as the direction of effect is similar in SP1 and SP3, even though the amplitude of effect is different in SP1 compared to SP3. The distance matrices show that, using the cost distance, the SP2 surface is away from SP1 and SP3 surfaces, whereas, using the Euclidean distance, SP2 is closer to the SP3 than to the SP1 surface. By applying a K-means clustering, which forces 2/3 clusters to be merged together, we can verify that, using the Euclidean distance, SP2 and SP3 are merged together and separated away from SP1 (sub-optimal clustering). In contrast, using the cost distance, SP1 and SP3 are properly merged together (optimal clustering). These results show that the cost distance will tend to group together real data surfaces such as those in SP1 and SP3.

The second simulation is devoted to test the superiority of the cost distance over the correlation distance (*Figure 1—figure supplement 4*; simulation 2). The correlation distance computed with the scipy.spatial.distance.correlation function in python (https://scipy.org/) is defined as:

$$(u,v) = 1 - \frac{(u - \acute{u})(v - \acute{v})}{\| \, |u - \acute{u}| \, \|_2 \, \| \, |v - \acute{v}| \, \|_2}$$

where $u$ and $v$ are the mean of the vector $u$ and $v$, respectively. In this simulation, the reference surface in the SPs is built in such a way that there is no variation of LFC values over time. For the reference surface in SP1, LFC values are 0, 2.2, 1.8, 2.2, and 1.8 for, respectively, 80, 92, 111, 140, and 175 CAG repeats. SP2 has the same global dynamics, but the small oscillations of LFC values around 2 are inverted, that is, the LFC values are 0, 1.8, 2.2, 1.8, and 2.2. The amplitude in SP3 is much smaller compared to SP1 and SP2. SP1 is thus most similar to SP2 than to SP3 as the small oscillation of amplitude (0.4) between SP1 and SP2 remains small compared to the global amplitude of the signal which, in average, is equal to 2. The distance matrices show that, using the correlation distance, SP1 is closer to SP3 than to SP2, whereas, using the cost distance, SP1 and SP2 are closer to each other than to SP3. By applying a K-means clustering, we can verify that, using the correlation distance, SP1 and SP3 are merged together and separated away from SP2 (sub-optimal clustering). In contrast, using the cost distance, SP1 and SP2 are properly merged together (optimal clustering). These results show that the cost distance will tend to group together real data surfaces such as those in SP1 and SP2.

Together, these two simulation studies suggest that the cost distance is an efficient metric to account for both a similar dynamic and similar amplitude of an effect.

## Temporal dynamics of cell-type-specific molecular responses

To discern the nature and temporal evolution of cell-type-specific molecular responses, we combined information from GDS cluster centroids and functional shRNA screen, the latter data in which the whole striatum of symptomatic HdhQ175 mice at 10 months of age was infected with shRNA pools using AAVs (AAV9) that preferentially target neurons and the effect on striatal cell survival was then measured via shRNA sequencing (**Wertz et al., 2020**). In this study, p<0.05 was used to retain shRNAs that enhance and suppress neuronal survival (**Wertz et al., 2020**). Herein, we used a less stringent threshold (p<0.1) to retain such shRNAs as this threshold enables a broader view on how molecular responses may develop in the striatum of Hdh mice on a system level, defining conclusions reached at high confidence (gene perturbation significant at p<0.05) or at a lower yet worth considering confidence (gene perturbation significant at 0.05<p<0.1). For every gene that belongs to a cell-type-specific cluster of GDS (centroid), a pathogenic effect was defined by statistically significant up-regulation of that gene accompanied by shRNA protection (sequence enrichment) or, alternatively, by a statistically significant down-regulation of that gene accompanied by shRNA-mediated cell death (sequence depletion), and vice versa for defining compensatory effects. Temporal subtypes of molecular responses were defined by considering the evolution of LFC values in Q175 mice at 2 months compared to 6 months of age and at 6 months compared to 10 months of age. A variation in the direction of molecular response was deemed to be significant if variations of LFC values represent at least 25% of the largest deregulation that is shown by the gene expression surface. This filter was applied to classifying the temporal subtypes of either pathogenic or compensatory responses (no change in the sign of LFC). For example, if the LFC values at 2, 6, and 10 months are 0, 0.5, and 1, respectively, the response is deemed to increase linearly over time because the difference between each LFC is >0.25. This filter was also applied to detecting transitions between pathogenic and compensatory responses (change in the sign of LFC). For example, if the LFC values at 2, 6, and 10 months are 0, –0.1, and 1, respectively, the response is deemed to show a biphasic profile. In each cell type, this analysis thus enabled to identify up to 10 temporal subtypes of molecular responses.

## Biological significance of *Geomic* data

To assess the biological significance of gene deregulation effects that are associated to specific striatal cell types (cell-type-specific GDS clusters) and to molecular response families as inferred from *Geomic* analysis, we subjected gene lists to STRING version 11.0 analysis (**Szklarczyk et al., 2017**; https://string-db.org/) in which stringent settings were used, namely information from 'Databases' and 'Experiments' only, a STRING confidence score ≥0.7 and 40–60 additional interactors allowed on the first shell. To assess the biological significance of molecular response categories, we applied the same approach, however allowing additional first-shell interactors in the range of greater than twice the number of seeds to a maximum of 80 interactors added. To assess network convergence of the increased pathogenic and lost compensatory responses, we allowed additional five times more first-shell interactors compared to the number of seeds, using a STRING confidence score ≥0.4 and information from 'Databases,' 'Experiments,' and 'co-expression.' In all types of analysis, the resulting networks were tested for the top annotations (smallest p-values for the largest numbers of nodes) provided by GO (KEGG pathways, biological processes). PubMed searches were also used to assess the biological significance of gene deregulation effects with a cellular assignment.

## Disease relevance of *Geomic* data

To assess the HD relevance of the genes implicated into molecular response families discerned by *Geomic* analysis, we tested for overlap between the mouse genes implicated into these responses and (i) human orthologs of mouse genes that modify neurobehavioral phenotypes in Drosophila (**Al-Ramahi et al., 2018**) and *C. elegans* (**Lejeune et al., 2012**) models of HD pathogenesis, (ii) SNP data about human orthologs that modify early-stage human disease in a total of 1991 human HD carriers, as measured using a composite progression score (**Moss et al., 2017**) or that modify the age-

at-onset of motor symptoms in over 9000 human HD patients (*Genetic Modifiers of Huntington's Disease (GeM-HD) Consortium, 2019*), (iii) human orthologs of mouse genes deregulated (q-value<0.01) in the caudate nucleus of the human prodromal HD brain (*Agus et al., 2019*), and (iv) HTT partner proteins identified from human, mouse, and rat libraries as listed in the HINT resource of the HdinHD database (https://www.hdinhd.org/). We performed these analyses for the genes implicated in the molecular responses defined at the cell type level (group I) and a larger group involving group I genes and the genes that modulate neuronal survival but for which cellular assignment is not available (group II). Orthologous genes were identified using the R package *biomaRt* (https://bioconductor.org/packages/release/bioc/html/biomaRt.html) and the DRSC integrative ortholog prediction tool (*Hu et al., 2011*), followed by manual curation. To calculate statistical significance, we performed hypergeometric tests in which the reference dataset for each comparison was the intersection between (i) all mouse genes with an ortholog and a cellular assignment (reference for group I), or all mouse genes with an ortholog and for which data on whole-striatum expression, cell-type-specific expression, and effect on striatal neuron survival are available in Hdh mice (reference for groupII) and (ii) all genes that were tested in the comparison dataset and that have a mouse ortholog. To generate Venn diagrams, we used the R packages *GeneOverlap* (https://bioconductor.org/packages/release/bioc/html/GeneOverlap.html) and *VennDiagram* (https://cran.r-project.org/web/packages/VennDiagram/index.html). An overlap was considered significant for p<0.05. Additionally, we performed PubMed searches, testing whether the genes that are involved in the molecular responses defined by *Geomic* analysis and that are strongly deregulated across CAG repeat and age conditions (maximum LFC>0.5) may modify HD pathogenesis, notably neuronal dysfunction phenotypes, across invertebrate, cellular, and murine models of the disease.

## Code availability

*Geomic* analysis can be adapted to any type of omic data that cover at least four points for a given variable. The source code developed for running *Geomic* analysis (Geomic package version 1.0), written using python, is available at http://www.broca.inserm.fr/geomic/index.php.

## Acknowledgements

This work was supported by Sorbonne Université, CNRS and INSERM, Paris, France, and by the CHDI Foundation (grant number A-14814), Princeton, USA (CN), and by CNRS-Momentum 2017 award (BG) and by grants from the CDHI Foundation, the JPB Foundation and NIH/NINDS (A-11582, PIIF, 1 R01 NS100802, and 1 R01 NS085880) (MH). We thank Pascal Frey and the ISCD (Sorbonne Université) for access to high-performance computing facilities. JC was the recipient of a PhD fellowship from the Systems Biology network (BioSys) of Sorbonne Université.

## Additional information

### Funding

| Funder | Grant reference number | Author |
| --- | --- | --- |
| CHDI Foundation | A-14814 | Christian Neri |
| Centre National de la Recherche Scientifique | Momentum 2017 | Barbara Gris |
| CHDI Foundation | A-11582 | Myriam Heiman |
| National Institutes of Health | R01 NS100802 | Myriam Heiman |
| JPB Foundation | PIIF | Myriam Heiman |

The funders had no role in study design, data collection and interpretation, or the decision to submit the work for publication.

### Author contributions

Lucile Megret, Software, Formal analysis, Methodology, Writing - original draft, Writing - review and editing; Barbara Gris, Software, Formal analysis, Writing - review and editing; Satish Sasidharan Nair,

Data curation, Visualization; Jasmin Cevost, Formal analysis; Mary Wertz, Resources, Data curation; Jeff Aaronson, Jim Rosinski, Thomas F Vogt, Hilary Wilkinson, Resources, Validation; Myriam Heiman, Resources, Data curation, Validation, Writing - review and editing; Christian Neri, Conceptualization, Formal analysis, Supervision, Methodology, Writing - original draft, Writing - review and editing

**Author ORCIDs**
Lucile Megret (iD) https://orcid.org/0000-0002-6835-6033
Christian Neri (iD) https://orcid.org/0000-0002-3790-2978

**Decision letter and Author response**
Decision letter https://doi.org/10.7554/eLife.64984.sa1
Author response https://doi.org/10.7554/eLife.64984.sa2

## Additional files

### Supplementary files
• Supplementary file 1. List of genes for which the whole-striatum gene deregulation in Hdh model mice is assigned to a specific cell type with false discovery rate (FDR) < 0.1. Cellular assignments are based on matching whole-striatum gene expression dysregulation curves to cell-type-specific gene expression dysregulation curves in the striatum of Hdh model mice at 6 months of age using the cost distance, after linear interpolation (see *Figure 1*: step 1). Data are shown for each cell type with indication of p-values and FDR values.

• Supplementary file 2. List of genes recruited in the clusters of gene deregulation surfaces and their centroids. Data are shown for each striatal cell type in Hdh model mice. Top biological annotations (KEGG pathways and gene ontology biological processes upon STRING database analysis; see Materials and methods) are also indicated for each gene deregulation surface cluster and each cell type.

• Supplementary file 3. List of genes underlying the molecular responses defined by the integration of cluster centroid data and in vivo functional data in the striatum of Hdh model mice. Data are shown for compensatory responses and for pathogenic responses, with indication of the evolution (increased then maintained, increased then reduced, transition) over time. The three columns labeled 'Information on cellular assignment' provides the cell type(s) to which gene deregulation in the whole striatum at 6 months is attributed, the weighted distance between whole-striatum and cell-type-specific log-fold-change (LFC) curves, and the false discovery rate (FDR) for the cellular assignments of gene deregulation. The six columns labeled 'Information on molecular response in Q175 mice' provide information on the functional effect of shRNA treatment (p-value and rank), the identity number of the gene deregulation surface (GDS) centroid to which the gene belongs, temporal subtype of molecular response, strength of mitigation (if applicable), and maximum deregulation of gene expression across age points.

• Transparent reporting form

### Data availability
Source data for RNA-seq analysis of whole striatum of Hdh mice (Langfelder et al. 2016) and functional shRNA screen data (score) in the striatum of Hdh mice (Wertz et al. 2020) have been previously reported. The visualization of whole-striatum RNA-seq data (Langfelder et al. 2016), cell type-specific gene expression data (LFC values) (Lee et al. 2020) and data generated by Geomic analysis including the nomination of cell types associated with whole-striatum gene expression dysregulation, cell-type-specific gene-deregulation-surface clusters (centroids) and the type and temporal evolution of molecular responses in specific cell types and their corresponding networks and biological annotations are available at http://www.broca.inserm.fr/geomic/index.php (this database might take some time to load at first consultation).

The following previously published datasets were used:

| Author(s) | Year | Dataset title | Dataset URL | Database and Identifier |
|---|---|---|---|---|
| Wertz MH, Mitchem MR, Pineda SS, Hachigian LJ, Lee H, Lau V, Powers A, Kulicke R, Madan GK, Colic M, Therrien M, Vernon A, Beja-Glasser VF, Hegde M, Gao F, Kellis M, Hart T, Doench JG, Heiman M | 2020 | Genome-wide In Vivo CNS Screening Identifies Genes that Modify CNS Neuronal Survival and mHTT Toxicity | https://www.ncbi.nlm.nih.gov/geo/query/acc.cgi?acc=GPL24247 | NCBI Gene Expression Omnibus, GPL24247 |
| Lee H, Fenster RJ, Pineda SS, Gibbs WS, Mohammadi S, Davila-Velderrain J, Garcia FJ, Therrien M, Novis HS, Gao F, Wilkinson H, Vogt T, Kellis M, LaVoie MJ, Heiman M | 2020 | Cell Type-Specific Transcriptomics Reveals that Mutant Huntingtin Leads to Mitochondrial RNA Release and Neuronal Innate Immune Activation | https://www.ncbi.nlm.nih.gov/geo/download/?acc=GSE152058 | NCBI Gene Expression Omnibus, GSE152058 |
| Langfelder P, Cantle JP, Chatzopoulou D, Wang N, Gao F, Al-Ramahi I, Lu XH, Ramos EM, El-Zein K, Zhao Y, Deverasetty S, Tebbe A, Schaab C, Lavery DJ, Howland D, Kwak S, Botas J, Aaronson JS, Rosinski J, Coppola G, Horvath S, Yang XW | 2016 | Integrated genomics and proteomics define huntingtin CAG length-dependent networks in mice | https://www.ncbi.nlm.nih.gov/geo/query/acc.cgi?acc=GPL13112 | NCBI Gene Expression Omnibus, GPL13112 |

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
