## [Decision Letter]

**Acceptance summary:**

A novel computational approach, *Geomic* is used to integrate three Huntington disease (HD) datasets and assess changes with disease progression in two pathways previously linked to disease; homeostatic and pathogenic responses in four striatal cell types in a HD model mice as well as human stem cell HD models. The data and analysis support the concept that a major driver of disease is loss of homeostatic pathways. The authors nicely address issues/concerns raised with previous submission.

**Decision letter after peer review:**

Thank you for submitting your article "Temporal dynamics of cell type-specific homeostatic and pathogenic responses to mutant huntingtin" for consideration by *eLife*. Your article has been reviewed by three peer reviewers, one of whom is a member of our Board of Reviewing Editors, and the evaluation has been overseen by Huda Zoghbi as the Senior Editor. The following individual involved in review of your submission has agreed to reveal their identity: Joan S Steffan (Reviewer #2).

The reviewers have discussed the reviews with one another and the Reviewing Editor has drafted this decision to help you prepare a revised submission.

Summary:

This work presents a novel computational approach, *Geomic*, and uses it to integrate three Huntington disease (HD) datasets to assess changes with disease progression in two pathways previously linked to disease; homeostatic and pathogenic responses in four striatal cell types in a HD model mice as well as human stem cell HD models. The data and analysis support the concept that a major driver of disease is loss of homeostatic pathways. The *Geomic* approach is likely to be of considerable use as a tool in the integration of large datasets.

Essential revisions:

1) The novelty of the *Geomic* method in comparison to existing methods needs to be more clearly presented/discussed.

2) A gene's cell type(s) expression is assigned to the entire bulk RNA-seq gene deregulation surface (GDS, i.e. the log fold changes across both time and number of repeats) by using the shape deformation method on just the data across expanding CAG repeats at just the six month timepoint. If that's correct, need to demonstrate better that you see the same expression pattern for an individual gene across expanding CAG repeats at the six month timepoint as you do across time with one CAG repeat.

3) What about cancellation effects, i.e. if the pattern in different cell types is drastically different and cancel each other out? For example, if a gene is upregulated in one cell type, downregulated in a second cell type at about the same abundance as the first type, and then a third cell type exhibits no change? *Geomic* can be adapted to handle cancellation events. Was it here? Why not?

---

## [Author Response]

Essential revisions:1) The novelty of the Geomic method in comparison to existing methods needs to be more clearly presented/discussed.

We have now presented the novelty of the *Geomic* method in comparison to existing methods in a clearer manner, both in the Introduction and at the beginning of the Discussion.

2) A gene's cell type(s) expression is assigned to the entire bulk RNA-seq gene deregulation surface (GDS, i.e. the log fold changes across both time and number of repeats) by using the shape deformation method on just the data across expanding CAG repeats at just the six month timepoint. If that's correct, need to demonstrate better that you see the same expression pattern for an individual gene across expanding CAG repeats at the six month timepoint as you do across time with one CAG repeat.

This is correct: a gene's cell type(s) expression is assigned to the entire bulk RNA-seq gene deregulation surface by using the shape deformation method on the data across expanding CAG repeats at the 6-month timepoint.

However, the assumption that expression pattern for an individual gene across expanding CAG repeats at the six-month timepoint is the same for that gene across time is not necessarily true as some genes are dysregulated in a linear manner across time and some others are not. In many instances, a gene that is downregulated across CAG repeats in a linear manner can be dysregulated across time in a non-linear (i.e. increase then decrease, or vice-versa) manner, providing the basis for the reduction of homeostatic responses or that of pathogenic responses over time.

To improve clarity, we have now added these considerations in the legend of Figure 1.

3) What about cancellation effects, i.e. if the pattern in different cell types is drastically different and cancel each other out? For example, if a gene is upregulated in one cell type, downregulated in a second cell type at about the same abundance as the first type, and then a third cell type exhibits no change? Geomic can be adapted to handle cancellation events. Was it here? Why not?

In the section on this topic, we now more clearly indicate that *Geomic* can be adapted to handle cancellation effects but that we did not perform this, as detecting cancellation effects does not enable a conclusion to be reached about the cellular assignment(s) of gene dysregulation (Discussion), and thus about the dynamics of the molecular response that is associated with gene dysregulation. However, we now discuss that cancellation effects could be indicative of a compensatory response in one cell type that is accompanied by a pathogenic response in another cell type. Additionally, we now provide 4 examples of gene expression patterns that clearly correspond to such cancellation effects (Discussion).